# Effect of Pregnancy on Postoperative Nausea and Vomiting in Female Patients Who Underwent Nondelivery Surgery: Multicenter Retrospective Cohort Study

**DOI:** 10.3390/ijerph192215132

**Published:** 2022-11-16

**Authors:** Jong-Ho Kim, Namhyun Kim, Soo-Kyung Lee, Young-Suk Kwon

**Affiliations:** 1Department of Anesthesiology and Pain Medicine, Chuncheon Sacred Heart Hospital, College of Medicine, Hallym University, Chuncheon-si 24253, Republic of Korea; 2Institute of New Frontier Research Team, College of Medicine, Hallym University, Chuncheon-si 24252, Republic of Korea; 3Department of Anesthesiology and Pain Medicine, Hallym University Sacred Heart Hospital, College of Medicine, Hallym University, Anyang 14068, Republic of Korea

**Keywords:** nausea, vomiting, anesthesia, surgery, operation

## Abstract

Pregnant women usually have several risk factors of postoperative nausea and vomiting (PONV) and physiologic changes that make them susceptible to PONV development. We investigated the risk of PONV and postoperative vomiting (PV) in pregnant women in nondelivery surgery compared to nonpregnant women. This study included female adult patients who underwent nondelivery surgery at five hospitals between January 2011 and March 2021. To identify the association between pregnancy and PONV, logistic regression was used to calculate the odds ratio and 95% confidence intervals (CIs), adjusting for covariates. A total of 60,656 (nonpregnant women = 57,363 and pregnant women = 3293) complete patient outcomes and perioperative data were eligible for analysis. Although there was no significant association between pregnancy and PONV, the risk of PV in the pregnant women was 3.9-fold higher (95% confidence interval (95% CI), 3.06–4.97) than in the nonpregnant women. In addition, increased pregnancy duration increased the risk of PV (odds ratio (95% CI), 1.05 (1.01–1.09)) and preoperative nausea, and vomiting increased the risk of PONV (odds ratio (95% CI), 2.68 (1.30–5.54)) and PV (odds ratio (95% CI), 4.52 (2.36–8.69)). Pregnancy increased the risk of PV in female patients who underwent nondelivery surgery, and pregnancy duration and preoperative nausea and vomiting also were associated with PONV or PV.

## 1. Introduction

Pregnant women may sometimes need surgeries or procedures under anesthesia. Surgery unrelated to pregnancy has been reported in about 0.7 to 1.6% of pregnant women [1,2]. In this population-based study of U.S. birth records, the ratio of cerclage, a commonly performed obstetric procedure, was 0.3 to 0.4% [3]. Postoperative nausea and vomiting (PONV) are common complications in surgery under anesthesia, and they can be no exception in pregnant women. Although nausea and vomiting are generally harmless to the mother and baby, severe or persistent PONV can adversely affect both the mother and the fetus [4]. If pregnant women are not supplied with food or liquid due to PONV, they may become dehydrated and lose weight [4,5], which may not provide sufficient nutrition to the fetus, so it is necessary to consider the risk of developing PONV in a pregnant woman’s surgery.

The major risk factors of PONV are female, nonsmoker, PONV or motion sickness history, and postoperative opioids, which are used to predict PONV through a simplified scoring system [6]. Considering that pregnant women usually do not smoke, pregnant women have at least two major risk factors. In addition, pregnant women have physiological changes during pregnancy, such as decreased lower esophageal sphincter tone and morning sickness, which are associated with nausea and vomiting [7,8]. These processes are likely to increase the risk of PONV. To the best of our knowledge, there was no PONV study to compare between nonpregnant women and pregnant women in nondelivery surgery. We hypothesized that pregnant women have a higher risk than nonpregnant women in nondelivery surgery and investigated the risk of pregnant women on PONV compared to nonpregnant women. In addition, we investigated the effects of preoperative nausea and vomiting, and pregnancy duration on PONV in pregnant women who underwent nondelivery surgery.

## 2. Materials and Methods

### 2.1. Ethical Approval/Informed Consent

This study was approved by the Clinical Research Ethics Committee of our institution (IRB No. 2022-05-012). We conducted the study in accordance with the ethics regulations based on the Helsinki Declaration. The study included vulnerable participants, such as pregnant women, but because it was a retrospective analysis of clinical data acquired in the treatment process that had already been completed, the informed consent of all clinical trial subjects was exempted from the research approval of the institution.

### 2.2. Data Sources, Study Design, and Setting

All data were obtained from the clinical data warehouse (CDW) of five hospitals. The CDW is a database of medical records, prescriptions, laboratory tests, and image examination results, containing over 10 years of outpatient and inpatient data. Data of patients can be searched and extracted based on prescriptions, examinations, and diagnoses, among other variables. The CDW can provide medical records in an unstructured format in addition to the patient test, transfusion, and drug administration records. We conducted this retrospective, unmatched, and matched cohort study from January 2011 to March 2021.

### 2.3. Participants

We included patients who underwent procedures under anesthesia, and the patients were extracted by anesthesia code. Anesthesia included general anesthesia, spinal anesthesia, epidural anesthesia, and nerve block under anesthesia care, except for local anesthesia, which is no code. The following patients were excluded:Male patientsFemale patients over 50 years of age (reproductive age [9])Patients under the age of 18Patients undergoing a Cesarean section or termination of pregnancyPatients who are unable to express nausea (patients who have undergone ventilator treatment or are unconscious after surgery)Patients who undergo surgery again within 24 h of surgery because it may be affected by overlapping surgery and anesthesiaPatients discharged 24 h before surgery because the observation period is 24 hPatients with missing values in their records.

Patients were included as eligible for analysis if they underwent multiple procedures yet did not undergo reoperation within 24 h.

### 2.4. Primary Outcomes

We determined the primary outcome as PONV and PV. If patients had a history of nausea and vomiting in their medical records within 24 h after surgery, we determined positive results regardless of the order of nausea and vomiting. The words that express PONV are summarized in Table A1 in Appendix A. PV was determined as vomiting occurring, if patients had a history of vomiting in their medical records within 24 h after surgery regardless of occurring nausea.

### 2.5. Variables

The presence or absence of pregnancy during surgery was evaluated as the primary exposure. The covariates included PONV risk factors with positive clinical evidence and alleviative factors. Age, body mass index, smoking status, surgery time, general anesthesia, use of volatile anesthetics and nitrous oxide for anesthesia, use of serotonin receptor antagonists and dexamethasone, preoperative fasting duration (hour), use of postoperative opioids, laparoscopic surgery, cholecystectomy, obstetric and gynecologic surgery, and an American Society of Anesthesiologists physical status were included. To consider changing medicine and surgical technique according to time, the surgery year was also included as a covariate. In addition, to investigate the effect of preoperative nausea and vomiting of pregnant women on postoperative nausea and vomiting, we included preoperative nausea and vomiting within 24 h before surgery. When the analytic subject was a pregnant woman, two variables, such as presence of multifetal pregnancy and pregnancy duration (weeks), were added.

### 2.6. Statistical Methods

Continuous data were presented as median and interquartile ranges due to skew, while categorical data were presented as frequencies and percentages. Mann–Whitney test and chi-square test were performed according to the type of data. Binary primary outcomes were summarized using frequencies and percentages. Unadjusted and adjusted odds ratios with 95% confidence intervals (CIs) were calculated using logistic regression.

The effects of pregnancy on PONV and PV were investigated using propensity score matching as a sensitivity analysis after controlling covariates. In the sensitivity analysis, we used a propensity score matching of 1 (nonpregnancy): 1 (pregnancy) without replacement. In the propensity score matching, the estimation algorithm was logistic regression, and the matching algorithm was the nearest neighbor. The covariates for matching included age, body mass index, surgery year, smoking status, surgery time, general anesthesia, use of volatile anesthetics and nitrous oxide for anesthesia, use of serotonin receptor antagonists and dexamethasone, preoperative fasting duration (hour), use of postoperative opioids, laparoscopic surgery, cholecystectomy, obstetric and gynecologic surgery, and an American Society of Anesthesiologists physical status of >2. The variables used for matching were evaluated for confounding factors using absolute standardized differences. After matching, the unadjusted and adjusted odds ratios of pregnancy for PONV and PV were calculated using logistic regression. In the subgroup analysis, we investigated the odds ratio of pregnancy duration and preoperative nausea and vomiting for developing PONV and PV in pregnant women.

The hypothesis testing was two-sided and applied Bonferroni correction due to two primary outcomes (alpha = 0.05/2 = 0.025). All statistical analyses and propensity score matching were performed using SPSS (version 26.0; IBM Corp., Armonk, NY, USA).

## 3. Results

### 3.1. Odds Ratios of Pregnancy for Developing PONV and PV

Among the 332,033 eligible cases, 271,377 were excluded, and 60,656 complete patient outcomes and perioperative data were eligible for analysis. The reasons for excluding patients, the number of excluded patients, and the number of patients are summarized in Figure 1. Pregnant women numbered 3293 and nonpregnant women numbered 57,363. PONV and PV were developed in 7842 (12.9%) and 1565 (2.6%) patients, respectively. The characteristics of PONV risk factors and covariates are summarized in Table 1. The unadjusted and adjusted odds ratios of pregnancy for developing PONV and PV in pregnant women who underwent surgery are summarized Table 2. The adjusted odds ratio of all variables for PONV and PV including pregnancy duration are summarized in Table A2 in Appendix A.

### 3.2. Odds Ratios of Pregnancy Duration and Preoperative Nausea and Vomiting for Developing PONV and PV in Pregnant Women Who Underwent Nondelivery Surgery

Among the 3299 pregnant women who underwent nondelivery surgery, PONV and PV developed in 200 (6.1%) and 170 (5.2%) patients, respectively. As additional covariates, 247 (7.5%) patients were multifetal pregnancy, and the median of pregnancy duration was 18 (interquartile range, 14–22) weeks. The unadjusted and adjusted odds ratios of increasing pregnancy duration and preoperative nausea and vomiting for developing PONV in pregnant women who underwent surgery are summarized in Table 3 and Table 4, respectively. The adjusted odds ratio of all variables for PONV and PV in pregnant women who underwent nondelivery surgery are summarized in Table A3 in Appendix A.

### 3.3. Odds Ratio of Pregnancy for Developing PONV and PV after Propensity Score Matching

After matching, there were 1008 each of nonpregnant patients and pregnant patients. After matching, the ASD of all covariates was <0.1, except for N_2_O (ASD = 0.23). The characteristics of the PONV risk factors and covariates before and after matching are summarized in Table A4 in Appendix A. PONV and PV developed in 62 (6.2%) and 26 (2.6%) nonpregnant patients, respectively. Among the pregnant patients, 69 (6.9%) and 52 (5.2%) developed PONV and PV, respectively. The unadjusted and adjusted odds ratios of preoperative nausea and vomiting for developing PONV are summarized in Table 5.

## 4. Discussion

In the present study, we evaluated the effect on PONV and PV development of pregnancy in nondelivery surgery using a retrospective cohort analysis. In addition, the effect of pregnancy duration and preoperative nausea and vomiting on PONV was also investigated. Although pregnancy did not significantly increase the risk of PONV compared to nonpregnancy, pregnancy had a 3.9- and 2.3-times higher risk of PV before and after propensity score matching than nonpregnancy, respectively. The risk of PV increased by 7% as the number of weeks of pregnancy increased, and preoperative zone vomiting increased the risk of PONV and PV by 2.7-times and 4.5-times, respectively.

In our results, pregnancy was associated with only PV and not PONV. This result is associated with the low incidence of nausea. In the pregnant women (6.1%) in our study, the incidence of PONV was not high compared with that in nonpregnant women (13.3%). There may be some causes for the low incidence of nausea. Some degree of nausea with or without vomiting develops in up to 90% of pregnancies [10]. Nausea is a common symptom during pregnancy, and pregnant women may underestimate nausea and complain less. Moreover, nausea is defined as a feeling of sickness or discomfort in the stomach that may come with an urge to vomit. Hence, it may be difficult to evaluate nausea objectively. In addition, charts may reveal fewer cases of PONV than the complaints of patients and direct and specific questions [6,11,12]. Vomiting is a more severe symptom than nausea. The incidence of PV was higher in pregnant women than in nonpregnant women. In pregnant women, vomiting without nausea may occur extensively, or nausea may be overlooked when nausea overlaps with vomiting.

The exact mechanism of nausea and vomiting during pregnancy is unknown. However, as it is widely accepted that gestational vomiting is caused by various metabolic and endocrine factors of placental origin [13], the mechanisms that increase the risk of PV in nondelivery surgery are probably multifactorial and more complex due to anesthesia and surgery. Pregnancy-related hormones, such as progesterone and estrogen, relax smooth muscle and thus slow the gastrointestinal transit time and may alter and/or delay gastric emptying [14,15]. If progesterone increases in pregnancy, the lower esophageal sphincter has more frequent and higher relaxation. As the uterus grows during the second and third trimesters of pregnancy, the pressure in the abdominal cavity increases, gradually increasing the pressure on the stomach [14]. These serial processes due to hormonal and physical changes may affect the development of PV and support our results that the risk of PONV and PV increases when pregnancy duration increased.

In data before matching, the use of volatile anesthetics and preventive antiemetics was less in pregnant women than in nonpregnant women. The difference in the use of volatile anesthetics between pregnant women and nonpregnant women may be associated with the difference in general anesthesia. For pregnant women, especially in most cases where neuraxial or general anesthesia is possible later in pregnancy, neuraxial anesthesia is primarily preferred to avoid the need for airway management, minimize fetal exposure to anesthetics, and reduce nausea, vomiting, and sedation compared with general anesthesia [16]. Moreover, the proportions of the use of 5-HT3 antagonists and dexamethasone were 31- and 27-fold higher in nonpregnant women than in pregnant women, respectively. Considering that almost all pregnant women were nonsmokers and 9.5% were exposed to postoperative opioids, pregnant women were at a medium or high risk of developing PONV. Despite the need for preventive multimodal intervention with a medium or high risk of PONV, pregnant women may not want to have them administered because 5-HT3 antagonists and dexamethasone are not indispensable in anesthesia and they are included in the FDA pregnancy categories B and C, respectively. According to a study on the perception of the risk of developing teratogenic effects in women with nausea and vomiting during pregnancy, 65.8% of women perceived that drug use for nausea and vomiting was more likely to increase the risk of birth defects in their babies [17]. However, except for some drugs, there is no clear evidence that any specific anesthesia-related agents are teratogenic in humans or that a specific anesthesia-related medication should be avoided during the perioperative care of a pregnant patient among anesthesia-related agents [2]. Even though drugs used to prevent or treat nausea and vomiting have no clear evidence of exerting a teratogenic effect on the fetus in the real field, anesthesiologists may experience difficulty in the prevention and treatment of PONV in pregnant women.

The strength of this study is that it is the first study investigating the effects of pregnancy on PONV in-patients who undergo nondelivery surgery. It was difficult to conduct the study, with pregnant women as the subjects, due to problems with ethics and informed consent. So far, the effects of pregnancy on PONV and PONV in studies of pregnant women have been unknown, and most studies involving pregnant women dealt with them having a Cesarean delivery [18,19,20,21,22,23]. However, this study has some limitations. First, because all data were obtained retrospectively, it was difficult to control all covariates. So, we included known risk factors for PONV and factors that could affect outcome in order to obtain adjusted results, and performed an analysis using propensity score matching in sensitivity analysis. Second, despite the importance of dose effects of N_2_O, antiemetics, and postoperative opioids on PONV, we used them as binary variables. It is well known that additional antiemetics are effective against PONV, and the use of N_2_O over 1 h and a high dose of postoperative opioids has effects on developing PONV. However, because our data concerning the history of PONV and details about medication, such as the timing, kind, number, and route of administration, were insufficient, we could not strictly control the covariates for analysis of the pregnancy effect on PONV. Additional antiemetics, use of N_2_O over 1 h, and high-dose opioids could affect the results, and it seems necessary to analyze the effects of the controlled use of medications in a further study. Third, it was not definitive with our data that preoperative nausea and vomiting were gestational nausea and vomiting. In general, a history of PONV and motion sickness is a risk factor for PONV [6]. Although the causes of nausea and vomiting due to pregnancy may be different with PONV and motion sickness, preoperative nausea and vomiting during pregnancy increased the risk of developing PONV. Although pregnant women had a lower incidence of preoperative nausea and vomiting than nonpregnant women, the risk of developing PONV during pregnancy was higher than during nonpregnancy. Further study is needed to investigate the association between gestational nausea and vomiting and PONV.

In conclusion, this study shows that pregnancy is not associated with PONV but with increased PV in nondelivery surgery. Pregnant women are more likely to be exposed to PONV risk factors than nonpregnant women, and it is more difficult to use drugs with them to prevent or treat PONV. Because vomiting is a more severe symptom than nausea, more thorough preparation is needed for prevention.

## Figures and Tables

**Figure 1 ijerph-19-15132-f001:**
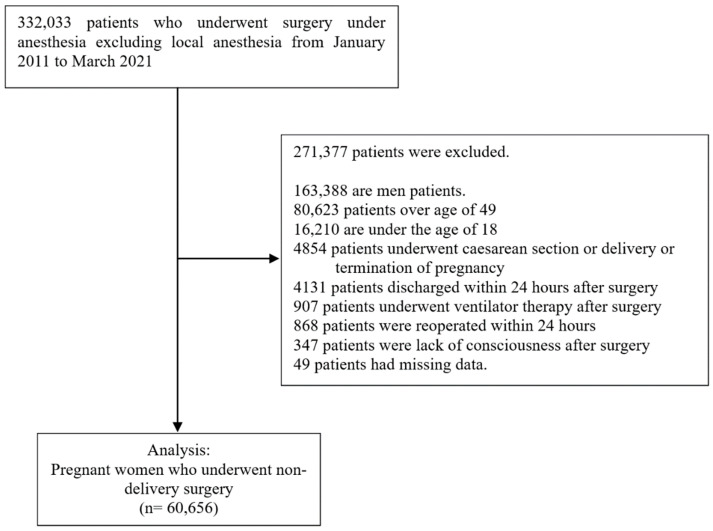
Flow chart.

**Table 1 ijerph-19-15132-t001:** Characteristics of PONV risk factors and covariates. Values are numbers (percentages) or medians (interquartile ranges).

	No Pregnancy (n = 57,363)	Pregnancy (n = 3293)	ASD
Age (years)	39 (31, 45)	33 (31, 36)	1.05
Body mass index (kg/m^2^)	22.7 (20.6, 25.5)	23.4 (21.3, 26.5)	0.21
Surgery year	2016 (2013, 2018)	2015 (2013, 2018)	0.04
Smoking	3544 (6.2)	9 (0.3)	1.13
Preoperative nausea and vomiting	2227 (3.9)	64 (1.9)	0.14
Preoperative fasting duration (hours)	11.5 (9, 13.9)	11.8 (9.9, 15)	3.32
Surgery time (hours)	1 (0.6, 1.8)	0.2 (0.1, 0.3)	0.41
ASA PS >2	1305 (2.3)	8 (0.2)	1.08
General anesthesia	51,546 (89.9)	1237 (37.6)	0.93
Volatile anesthetics	47,350 (82.5)	1233 (37.4)	3.36
N_2_O	6792 (11.8)	4 (0.1)	3.84
Serotonin antagonists	28,620 (49.9)	53 (1.6)	1.08
Dexamethasone	1563 (2.7)	2 (0.1)	0.2
Postoperative opioids	44,792 (78.1)	313 (9.5)	2.34
Laparoscopic surgery	19,978 (34.8)	39 (1.2)	3.11
Cholecystectomy	3566 (6.2)	3 (0.1)	2.03
Obstetric and gynecologic surgery	18,538 (32.3)	3236 (98.3)	5.06

PONV, postoperative nausea and vomiting; ASD, absolute standardized differences; ASA, American Society of Anesthesiologists; PS, physical status.

**Table 2 ijerph-19-15132-t002:** Odds ratio of pregnancy on developing PONV and PV in female who underwent nondelivery surgery.

	PONV		PV	
	Odds Ratio (95% CI)	*p* Value	Odds Ratio (95% CI)	*p* Value
Unadjusted	0.42 (0.36–0.49)	<0.001	2.18 (1.86–2.57)	<0.001
Adjusted	0.92 (0.77–1.09)	0.327	3.9 (3.06–4.97)	<0.001

PONV, postoperative nausea and vomiting; PV, postoperative vomiting. The adjusted odds ratio is the odds ratio of pregnancy duration in a multivariate analysis with other variables (age, body mass index, surgery year, smoking, preoperative nausea and vomiting, preoperative fasting duration, surgery time, American Society of Anesthesiologists physical status, general anesthesia, use of volatile anesthetics and N_2_O, use of serotonin antagonists and dexamethasone, use of postoperative opioids, laparoscopic surgery, cholecystectomy, obstetric and gynecologic surgery, and multifetal pregnancy) included.

**Table 3 ijerph-19-15132-t003:** Odds ratio of pregnancy duration for developing PONV and PV in pregnant women who underwent nondelivery surgery.

	PONV		PV	
	Odds Ratio (95% CI)	*p* Value	Odds Ratio (95% CI)	*p* Value
Unadjusted	1.00 (0.97–1.0.3)	0.855	1.07 (1.03–1.10)	<0.001
Adjusted	0.99 (0.96–1.03)	0.747	1.05 (1.01–1.09)	0.015

PONV, postoperative nausea and vomiting; PV, postoperative vomiting. The adjusted odds ratio is the odds ratio of pregnancy duration in a multivariate analysis with other variables (age, body mass index, surgery year, smoking, preoperative nausea and vomiting, preoperative fasting duration, surgery time, American Society of Anesthesiologists physical status, general anesthesia, use of volatile anesthetics and N_2_O, use of serotonin antagonists and dexamethasone, use of postoperative opioids, laparoscopic surgery, cholecystectomy, obstetric and gynecologic surgery, and multifetal pregnancy) included.

**Table 4 ijerph-19-15132-t004:** Odds ratio of preoperative nausea and vomiting for developing PONV and PV in pregnant women who underwent nondelivery surgery.

	PONV		PV	
	Odds Ratio (95% CI)	*p* Value	Odds Ratio (95% CI)	*p* Value
Unadjusted	2.69 (1.49–5.91)	0.002	5.52 (2.99–10.19)	<0.001
Adjusted	2.68 (1.30–5.54)	0.008	4.52 (2.36–8.69)	<0.001

PONV, postoperative nausea and vomiting; PV, postoperative vomiting. The adjusted odds ratio is the odds ratio of pregnancy in a multivariate analysis with other variables (age, body mass index, surgery year, smoking, preoperative fasting duration, surgery time, American Society of Anesthesiologists physical status, general anesthesia, use of volatile anesthetics and N_2_O, use of serotonin antagonists and dexamethasone, use of postoperative opioids, laparoscopic surgery, cholecystectomy, obstetric and gynecologic surgery, pregnancy duration, and multifetal pregnancy) included.

**Table 5 ijerph-19-15132-t005:** Odds ratio for pregnancy on developing PONV and PV in reproductive female patients who underwent nondelivery surgery after propensity score matching.

	PONV		PV	
	Odds Ratio (95% CI)	*p* Value	Odds Ratio (95% CI)	*p* Value
Unadjusted	0.53 (0.79–1.60)	0.527	2.05 (1.27–3.32)	0.003
Adjusted	1.22 (0.84–1.75)	0.295	2.27 (1.37–3.75)	0.001

PONV, postoperative nausea and vomiting; PV, postoperative vomiting. The adjusted odds ratio is the odds ratio of pregnancy in a multivariate analysis with other variables (age, body mass index, surgery year, smoking, preoperative nausea and vomiting, preoperative fasting duration, surgery time, American Society of Anesthesiologists physical status, general anesthesia, use of volatile anesthetics and N_2_O, use of serotonin antagonists and dexamethasone, use of postoperative opioids, laparoscopic surgery, cholecystectomy, and obstetric and gynecologic surgery) included.

## Data Availability

Restrictions apply to the availability of these data. Data were obtained from Hallym Medical Center and are available from the clinical data warehouse of Hallym Medical Center with permission.

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
