# Peer review of "Effect of Pregnancy on Postoperative Nausea and Vomiting in Female Patients Who Underwent Nondelivery Surgery: Multicenter Retrospective Cohort Study"

_ijerph, 2022, doi:10.3390/ijerph192215132_

Round 1

Reviewer 1 Report

Dear Editor,

Thank you for the opportunity to review the paper by Kim et al about the association of pregnancy on PONV in non-delivery surgery.

 I have some comments I hope the authors are able to adress.

Please can the authors explain why they report p-values in the descriptives table 1, appendix table 2 and appendix table 3? What is the null hypothesis here? Significance tests should be avoided in descriptive tables. Can the authors explain why they think that the p-value is relevant in this setting? I can recommend reading Altmans paper about p-values from 2016: Statistical tests, P values, confidence intervals, and power: a guide to misinterpretations. Eur J Epidemiol. 2016; 31: 337–350. doi:  10.1007/s10654-016-0149-3

Page 4 line 131: “Pregnant women were 57,363 and non-pregnant women were 3293.”Please rephrase this sentence. I assume it is the other way around as shown in Table 1?

Chapter 2.3 Please explain systematically inclusion and exclusion criteria. According to the flow scheme, local anesthesia was an exclusion criteria. And there were more too (Age, ventilator therapy and so on) this should be presented here.

Why are pregnant woman under the age of 18 excluded?

4259 patients were excluded due to lack of consciousness. I assume that these are female patients between the age of 18-49 (since all the others were already excluded) Why do you have so many patients in this age group that are unconsciousness after surgery? These are more patients than pregnant woman and the authors should explain this to the reader.

The authors analyse the association with pregnancy and PONV. But there are many confounding factors: Non-pregnant patients are getting more volatile anaesthesia than pregnant once, They receive more seretoninantagonists and more opioids. This need to be discussed.

Table 2, 3,  4 and 5: Please explain to the readers what you have adjusted for in the Table description.

The discussion section must be revised. It is difficult to read and the discussion would benefit from an English revision. Many sentences do not make sense.

The discussion is very long and yet does not focus properly on the key findings and issues. A robust re-appraisal of the strengths and weaknesses of the study is essential. The usual approach to the discussion would be 1) summarise key findings 2) discuss findings in context of existing literature 3) strengths and weaknesses 4) conclusions and suggestions for future work. This standard approach is helpful for the reader and much more informative.

Author Response

Reviewer 1

Comment 1.

Please can the authors explain why they report p-values in the descriptives table 1, appendix table 2 and appendix table 3? What is the null hypothesis here? Significance tests should be avoided in descriptive tables. Can the authors explain why they think that the p-value is relevant in this setting? I can recommend reading Altmans paper about p-values from 2016: Statistical tests, P values, confidence intervals, and power: a guide to misinterpretations. Eur J Epidemiol. 2016; 31: 337–350. doi:  10.1007/s10654-016-0149-3

Answer 1.

Thank you for your comments. I agree your opinion. Our null hypothesis is that pregnancy does not increase risk of PONV. I removed the p-value that is not related to the null hypothesis.

Comment 2.

Page 4 line 131: “Pregnant women were 57,363 and non-pregnant women were 3293.”Please rephrase this sentence. I assume it is the other way around as shown in Table 1?

Answer 2.

Thank you for your comments. I corrected it.

Pregnant women were 3293 and non-pregnant women were 57,363.

Comment 3.

Chapter 2.3 Please explain systematically inclusion and exclusion criteria. According to the flow scheme, local anesthesia was an exclusion criteria. And there were more too (Age, ventilator therapy and so on) this should be presented here.

Answer 3.

Thank you for your comments. I corrected it.

We included patients who underwent procedures under anesthesia, and the patients were extracted with anesthesia code. Anesthesia included general anesthesia, spinal anesthesia, epidural anesthesia and nerve block under anesthesia care, except for local anesthesia which is no code. The following patients were excluded:

  1. Male patients
  2. Female patients over 50 years of age (reproductive age [9])
  3. Patients under the age of 18
  4. Patients undergoing a cesarean section or termination of pregnancy
  5. Patients who are unable to express nausea (patients who have undergone ventilator treatment or are unconscious after surgery)
  6. Patients who undergo surgery again within 24 hours of surgery because it may be affected by overlapping surgery and anesthesia
  7. Patients discharged 24 hours before surgery because the observation period is 24 hours
  8. Patients with missing values in their records.

Comment 4.

Why are pregnant woman under the age of 18 excluded?

Answer 4.

Thank you for your opinion. I think that's a very good opinion, too. However, there were some restrictions. First of all, there were only four patients under the age of 18. All four of the total pregnant women were included in the exclusion criteria because they had undergone cesarean section or abortion. Also, our study was authorized only for adult patients (18 years of age or older).

Comment 5.

4259 patients were excluded due to lack of consciousness. I assume that these are female patients between the age of 18-49 (since all the others were already excluded) Why do you have so many patients in this age group that are unconsciousness after surgery? These are more patients than pregnant woman and the authors should explain this to the reader.

Answer 5.

Thanks for the good comments. Due to the difference in the order of exclusion, there appeared to be many unconscious patients after surgery. The number of unconscious patients included some men and some women under the age of 18 or over 50. We corrected the order of exclusion. There is no difference in total participants.

Figure 1. flow chart

Comment 6.

The authors analyse the association with pregnancy and PONV. But there are many confounding factors: Non-pregnant patients are getting more volatile anaesthesia than pregnant once, They receive more seretoninantagonists and more opioids. This need to be discussed.

Answer 6.

Thanks for the good comments. I added discussion about above contents.

The difference in the use of volatile anesthetics between pregnant women and nonpregnant women may be associated with the difference in general anesthesia. For pregnant women, especially in most cases where neuraxial or general anesthesia is possible later in pregnancy, neuraxial anesthesia is primarily preferred to avoid the need for airway management, minimize fetal exposure to anesthetics, and reduce nausea, vomiting, and sedation compared with general anesthesia. [16]. Moreover, the proportions of the use of 5-HT3 antagonists and dexamethasone were 31- and 27-fold higher in nonpregnant women than in pregnant women, respectively. Considering that almost all pregnant women were nonsmokers and 9.5% were exposed to postoperative opioids, pregnant women were at medium or high risk of developing PONV. Despite the need for preventive multimodal intervention in medium or high risk on PONV, pregnant women may not want to have administered them because 5-HT3 antagonists and dexamethasone are not indispensable in anesthesia and those are included in FDA pregnancy category B and C, respectively. According to a study on the perception of the risk of developing teratogenic effects in women with nausea and vomiting during pregnancy, 65.8% of women perceived that drug use for nausea and vomiting was more likely to increase the risk of birth defects in their babies [17]. However, except for some drugs, there is no clear evidence that any specific anesthesia-related agents are teratogenic in humans or that a specific anesthesia-related medication should be avoided during the perioperative care of a pregnant patient among anesthesia-related agents [2]. Even though drugs used to prevent or treat have no clear evidence of exerting a teratogenic effect on the fetus in the real field, anesthesiologists may undergo difficulty in the prevention and treatment of PONV in pregnant women.

Comment 7.

Table 2, 3,  4 and 5: Please explain to the readers what you have adjusted for in the Table description.

Answer 8.

Thank you for your comments. I have added description to tables 2,3,4 and 5.

Comment 8.

The discussion section must be revised. It is difficult to read and the discussion would benefit from an English revision. Many sentences do not make sense.

Answer 8.

Thank you for your comments. This article was revised by professional English correction and translation service provider.

Comment 9.

The discussion is very long and yet does not focus properly on the key findings and issues. A robust re-appraisal of the strengths and weaknesses of the study is essential. The usual approach to the discussion would be 1) summarise key findings 2) discuss findings in context of existing literature 3) strengths and weaknesses 4) conclusions and suggestions for future work. This standard approach is helpful for the reader and much more informative.

Answer 9.

Thank you for your comments. I revised according to your advice.

Reviewer 2 Report

No comment for the authors

Author Response

Thank you.

Reviewer 3 Report

Article needs improvement with respect to English language usage.

Author Response

Comment 1.

Article needs improvement with respect to English language usage.

Answer 1.

Thank you for your comments. This article was revised by professional English correction and translation service provider.

Round 2

Reviewer 1 Report

Dear Editor,

the authors performed a point-by-point rebuttal and the manuscript improved significantly. I have no further comment on the scientific content of this paper.